# Migration and Invasion Enhancer 1 Is an NF-ĸB-Inducing Gene Enhancing the Cell Proliferation and Invasion Ability of Human Prostate Carcinoma Cells In Vitro and In Vivo

**DOI:** 10.3390/cancers11101486

**Published:** 2019-10-02

**Authors:** Kang-Shuo Chang, Ke-Hung Tsui, Yu-Hsiang Lin, Chen-Pang Hou, Tsui-Hsia Feng, Horng-Heng Juang

**Affiliations:** 1Department of Anatomy, College of Medicine, Chang Gung University, Kwei-Shan, Tao-Yuan 33302, Taiwan; D0501301@stmail.cgu.edu.tw; 2Institute of Biomedicine, College of Medicine, Chang Gung University, Kwei-Shan, Tao-Yuan 33302, Taiwan; 3Department of Urology, Chang Gung Memorial Hospital, Kwei-Shan, Tao-Yuan 33302, Taiwan; t2130@cgmh.org.tw (K.-H.T.); laserep@mail.cgu.edu.tw (Y.-H.L.); glucose1979@gmail.com (C.-P.H.); 4Graduate Institute of Clinical Medical Science, College of Medicine, Chang Gung University, Kwei-Shan, Tao-Yuan 33302, Taiwan; 5School of Nursing, College of Medicine, Chang Gung University, Kwei-Shan, Tao-Yuan 33302, Taiwan; thf@mail.cgu.edu.tw

**Keywords:** prostate, MIEN1, Akt, NF-κB, IL-6, NDRG1, tumorigenesis, JSH-23, MK2206

## Abstract

Migration and invasion enhancer 1 (MIEN1) is a membrane-anchored protein and exists in various cancerous tissues. However, the roles of MIEN1 in prostate cancer have not yet been clearly addressed. We determined the expression, biological functions, and regulatory mechanisms of MIEN1 in the prostate. The results of immunohistochemical analysis indicated that MIEN1 was expressed specifically in epithelial cells and significantly higher in adenocarcinoma as compared to in normal tissues. MIEN1 enhanced in vitro cell proliferation, invasion, and in vivo tumorigenesis. Meanwhile, MIEN1 attenuated cisplatin-induced apoptosis in PC-3 cells. Overexpression of NF-ĸB-inducing kinase (NIK) enhanced MIEN1 expression, while overexpression of NF-ĸB inhibitor α (IĸBα) blocked MIEN1 expression in PC-3 cells. In prostate carcinoma cells, MIEN1 provoked Akt phosphorylation; moreover, MIEN1 downregulated N-myc downstream regulated 1 (NDRG1) but upregulated interleukin-6 (IL-6) gene expression. MK2206, an Akt inhibitor, impeded the modulation of MIEN1 on NDRG1 and IL-6 expressions. Our studies suggest that MIEN1 is an NF-ĸB downstream oncogene in the human prostate. Accordingly, the modulation of Akt signaling in the gene expressions of NDRG1 and IL-6 may account for the functions of MIEN1 in cell proliferation, invasion, and tumorigenesis in prostate carcinoma cells.

## 1. Introduction

Migration and invasion enhancer 1 (MIEN1/C35/C17orf37), as an oncogenic gene, is located on chromosome 17q12 and 505 nucleotides from the 3′ end of the ERBB2 oncogene [1,2]. Several studies have shown that MIEN1 overexpression results in either disease progression or metastatic propensity in many tumor types, including colon, prostate, ovarian, non-small cell lung, stomach, and oral cancers [3,4,5,6,7,8,9]. In addition, studies found that MIEN1 is overexpressed in various carcinomas and abundantly in distant metastases while it is expressed at low levels or zero in normal human tissues, suggesting that MIEN1 overexpression represents an oncogenic nature in the promotion of tumor cell dissemination and metastasis in breast cancer in vitro and in vivo [7,10]. However, the latest research stated that knockdown of the MIEN1 gene did not alter the morphology, development, proliferation, or survival of breast cancer cells [11]. 

A previous study disclosed that MIEN1 was expressed in the cytosol with intense staining in the membrane of prostate carcinoma cells, and it functionally enhanced migration and invasion via NF-κB/Akt activity [12]. Meanwhile, it was found that a novel miRNA, has-miR-940 (miR-940), reduced the anchorage-independent development capacity and enhanced mesenchymal-to-epithelial transition via the blockage of MIEN1 expression by inhibiting the migratory and invasive potential of prostate carcinoma cells [13]. Further study indicated that hypomethylation of the SINE Alu region of the MIEN1 promoter and upstream stimulatory factor 1 (USF1) binding triggered MIEN1 expression in prostate cancer in vitro and in vivo [6]. 

A study has suggested that MIEN1 may have an important regulatory role in the phosphorylation of Akt with its redox potential [14]. The downregulation of MIEN1 suppressed matrix metallopeptidase 9 (MMP9) expression by downregulating Akt expression, indicating that MIEN1 is a prospective molecular target for breast cancer chemotherapy [3]. Interleukin-6 (IL-6), a pluripotency cytokine, is involved in the malignant progression of prostate cancer [15,16], while N-myc downstream regulated 1 (NDRG1) is a tumor suppressor gene in numerous cancer cells [17], including prostate [18,19]. It has been suggested that both IL-6 and NDRG1 are modulated by Akt signal pathways [20,21]; however, no report has yet studied the correlations among MIEN1, Akt, IL-6, and NDRG1 in human prostate cancer. 

Although MIEN1 is regarded as a novel gene involved in prostate cancer progression by enhancing cell migration and invasion, the relative functional significance and regulatory mechanisms of MIEN1 remain largely unknown. In this research, we identified the MIEN1 expression levels in prostate carcinoma cells and normal prostate tissues and examined the potential functions and regulatory mechanisms of downstream MIEN1 target genes in prostate carcinoma cells. 

## 2. Results

### 2.1. Expression of MIEN1 in Prostate Carcinoma Cells and Prostate Tissues 

The protein levels of MIEN1 (Figure 1A) and mRNA (Figure 1B) in five cell lines of cultured prostate cells were compared. The results of immunoblot and RT-qPCR assays revealed that the highest protein levels of MIEN1 were detected in LNCaP cells among the five prostate cell lines. The prostate carcinoma cells (LNCaP, PC-3, and DU145) expressed higher levels of MIEN1 than did the normal or non-metastatic prostate cells (PZ-HPV-7 and CA-HPV-10). The immunohistochemical assay (IHC) results of a human prostate tissue array are demonstrated in Figure 1C. The immunostaining for MIEN1 in normal (*n* = 9), grade II (*n* = 8), and grade III (*n* = 32) prostate cancer tissues was determined by immunohistochemistry assays in a human prostate tissue array. The intense scores of MIEN1 immunostaining in normal prostate tissues were significantly lower than those in high-grade prostate cancer tissues, although the results showed no significant differences between grade II and grade III prostate cancer tissues (Figure 1D).

### 2.2. MIEN1 Is the Downstream Gene of NF-ĸB Signaling and Induces Akt Phosphorylation in Prostate Carcinoma Cells

Both NF-κB (NFluc) and MIEN1 reporter vectors transiently cotransfected with IκBα expression vector (pCMVIκBα) blocked the reporter activities of NF-κB and MIEN1, while with an NFκB-inducing kinase (NIK) expression vector (pcDNA-NIK) they upregulated reporter activity, as indicated by reporter assays (Figure 2A). Further results of the reporter assays showed that MIEN1 reporter vector cotransfected with various dosages of pCMVIκBα expression vectors downregulated MIEN1 reporter activity, while with pcDNA-NIK, it upregulated human MIEN1 promoter activity in a dose-dependent manner (Figure 2B). Immunoblot (Figure 2C) and RT-qPCR (Figure 2D) assays also revealed that MIEN1 expression was upregulated by overexpression of NIK but downregulated by IĸBα overexpression. The results of immunoblot assays showed that overexpression of MIEN1 in PC-3 (PC-MIEN1) cells induced Akt phosphorylation, while MIEN1-knockdown PC-3 (PC_shMIEN1) and MIEN1-knockdown LNCaP (LN_shMIEN1) cells had lower Akt phosphorylation in comparison to the mock-transfected cells (Figure 2E). Further immunoblot assays showed that treatment with MK2206, an Akt inhibitor, in MIEN1-overexpressed PC-3 cells attenuated the induction by MIEN1 of either Akt phosphorylation or protein levels of MMP9 (Figure 2F).

### 2.3. MIEN1 Enhances Cell Growth of Prostate Carcinoma Cells In Vitro and In Vivo

The [^3^H]thymidine incorporation assays revealed that knockdown of MIEN1 attenuated cell proliferation in LNCaP (Figure 3A) and PC-3 (Figure 3B) cells. The percentage of positive cells with 5-Ethynyl-2´-deoxyuridine (EdU) staining was increased when MIEN1 was overexpressed in PC-3 cells, as determined by flow cytometry (Figure 3C). Similar results were found in a colony formation assay as well. Clone numbers increased about 3 times when MIEN1 was overexpressed in PC-3 cells (Figure 3D), while MIEN1 knockdown in PC-3 cells inhibited colony formation (Figure 3E). Further study using xenografts in BALB/cAnN-Foxn1^NU^ mice revealed that tumors generated from MIEN1-knockdown PC-3 (PC_shMIEN1) cells grew slower than did those derived from mock-transduced PC-3 (PC_shCOL) cells (Figure 3F). Additionally, tumors generated from PC_shMIEN1 cells were approximately 26.85% smaller than the tumors generated from PC_shCOL cells (58.66 ± 25.59 vs. 218.48 ± 33.24 mm^3^) after 39 days of growth (Figure 3F,G). The tumor weight was significantly decreased in the PC_shMIEN1 group compared to in the PC_shCOL group (Figure 3H). Further immunoblot assays confirmed that MIEN1 was knocked down in the xenograft tumors by which PC_shMIEN1 cells were inoculated (Figure 3I). Collectively, knockdown of MIEN1 attenuated the cell proliferation and tumorigenesis of prostate carcinoma cells.

### 2.4. Ectopic Overexpression of MIEN1 Attenuates Cisplatin-Induced Cell Apoptosis in PC-3 Cells

Flow cytometric analysis with double fluorescence staining (Annexin V-FITC/PI) showed that PC-MIEN1 cells were significantly less apoptotic than PC-DNA cells after 24 h of 80 µM cisplatin treatment (Figure 4A). Figure 4B shows the results of quantitative analysis of the apoptotic cells after cisplatin treatment (28.57 ± 2.52 vs. 13.13 ± 1.04). Furthermore, the results of MTS assays indicated that cisplatin treatment decreased cell viability dose-dependently and that PC-MIEN1 cells had much greater resistance to cisplatin treatment (Figure 4C).

### 2.5. Effects of MIEN1 on Cell Invasion and Migration in Prostate Carcinoma Cells

Matrigel invasion assays revealed that PC-MIEN1 cells displayed markedly greater invasive capacity than did PC-DNA cells (Figure 5A). Moreover, the invasion capacity of PC-3 cells was reversed when MIEN1 was knocked down (Figure 5B). We also transiently transfected the human MIEN1 expression vector into prostate carcinoma DU145 cells. The results of immunoblot (Figure 5C, top) and RT-qPCR (Figure 5C, bottom) assays confirmed the two clones of ectopic overexpression of MIEN1 in DU145 (DU-MIEN1-1, DU-MIEN1-2) cells compared to in mock-transfected DU145 (DU-DNA) cells. In comparison to DU-DNA cells, the Matrigel invasion assays indicated that overexpression of MIEN1 promoted the invasion capacity of DU145 cells (Figure 5D). Further wound healing assays clearly presented overexpression of MIEN1 enhancing the migration ability of PC-3 cells in vitro (Figure 5E). Further results of F-actin staining indicated that MIEN1 ectopic overexpression increased the F-actin staining intensity at the leading edge of cells, while MEIN1 knockdown reversed this phenomenon (Appendix A). The RT-qPCR (Appendix A) and immunoblot (Appendix A) assays showed that ectopic overexpression of MIEN1 downregulated E-cadherin but upregulated N-cadherin and Slug gene expressions.

### 2.6. Regulation between MIEN1 and IL-6 in Prostate Carcinoma PC-3 Cells

Overexpression of MINE1 induced IL-6 expression as determined by RT-qPCR (Figure 6A) and enzyme-linked immunosorbent assay (ELISA) (Figure 6C). Moreover, expression of IL-6 was downregulated when MIEN1 was knocked down in PC-3 cells (Figure 6B,D). Reporter assays also showed that transient-cotransfected MIEN1 expression vector enhanced the reporter activity of IL-6 reporter vector (Figure 6E). The increased activity by MIEN1 of IL-6 expression in PC-3 cells was attenuated by MK2206 treatments, as determined by RT-qPCR (Figure 6F) and reporter (Figure 6G) assays. Interestingly, the results of the immunoblot, RT-qPCR (Figure 6H), and reporter (Figure 6I) assays revealed that IL-6 positively modulated MIEN1 gene expression. Treatments with JSH-23, an NF-κB inhibitor, not only downregulated the MIEN1 reporter activity but also blocked the activation of IL-6 on MIEN1 reporter activity (Figure 6J), indicating that IL-6 may upregulate MIEN1 gene expression via the NF-ĸB signaling pathway. 

### 2.7. MIEN1 Downregulates NDRG1 Gene Expression in Prostate Carcinoma Cells

The results of the immunoblot (Figure 7A, top) and RT-qPCR (Figure 7A, bottom) assays showed that overexpression of MIEN1 decreased NDRG1 expression, while knockdown of MIEN1 enhanced NDRG1 expression in PC-3 cells. Further immunoblot (Figure 7B, top) and RT-qPCR (Figure 7B, bottom) assays revealed that MIEN1-knockdown LNCaP (LN_shMIEN1) cells expressed higher levels of NDRG1 in comparison to mock-knockdown LNCaP (LN_shCOL) cells. However, the results of the RT-qPCR assays (Figure 7B, bottom) showed that knockdown of MIEN1 did not affect the expression of prostate-specific antigen (PSA) in LNCaP cells. The reporter assays showed that transient-cotransfected MIEN1 expression vector decreased the reporter activity of NDRG1 reporter vector in prostate carcinoma PC-3 and LNCaP cells (Figure 7C). Moreover, the results of the immunoblot (Figure 7D, top) and RT-qPCR (Figure 7D, bottom) assays indicated that treatment with MK2206 did not affect NDRG1 expression in PC-3 cells but attenuated the decreasing levels of NDRG1 expression by MIEN1.

## 3. Discussion 

The migration and invasion enhancer 1 (MIEN1/C35/C17orf37) gene is close to the oncogene ERBB2 and has been considered an oncogene in breast cancer [1,10,22]. Several reports from independent laboratories have shown that overexpression of MIEN1 induces metastatic potential and disease progression in many types of tumors [3,4,5,6,7,8,9]. The results of this study revealed that the levels of MIEN1 protein (Figure 1A) and mRNA (Figure 1B) in prostate carcinoma cells (LNCaP, PC-3, and DU145) were higher than those in normal prostate (PZ-HPV-7 and CA-HPV-10) cells. Although our study revealed that MIEN1 expression among the prostate carcinoma cell lines was dependent on the cell type but not relevant to the extent of neoplasia in vitro, further IHC analysis of a human prostate tissue array indicated that MIEN1 immunostaining in normal prostate tissues was less significant than that in high-grade prostate cancer tissues (Figure 1). Our findings suggest that MIEN1 is an oncogene in prostate cancer, and this is consistent with our prior research [6,12,13]. 

Recent studies have indicated that microRNAs target MIEN1 to block the migration and invasion ability of cancerous cells. miR-26b decreased MIEN1 levels in non-small-cell lung cancer cells, which downregulated the metastasis activity via the NF-κB/MMP-9/VEGF (vascular endothelial growth factor) pathways [7]. Further, miR-940 modified MIEN1 RNA to inhibit the migratory and invasive potential of cancer cells, attenuating their anchorage-independent growth capacity [13]. However, the regulatory mechanisms of human MIEN1 at the transcriptional levels are still not well defined. Our study is the first report indicating NF-κB-modulated gene expression of MIEN1 in human prostate carcinoma cells. Ectopic IκBα overexpression downregulated MIEN1 expression, while NFκB-inducing kinase (NIK) overexpression upregulated MIEN1 expression, suggesting that MIEN1 is the target gene of the NF-κB signaling pathway. Interestingly, the findings of this research verified that MIEN1 causes Akt phosphorylation. Besides this, MK2206, an Akt inhibitor in MIEN1-overexpressing PC-3 cells, reduced MIEN1 activation in Akt phosphorylation and MMP9 protein concentrations (Figure 2). These results on MIEN1 and the Akt signaling pathway are in agreement with recent studies [3,4,14,23]. 

Most studies on MIEN1 have focused on migration and invasion; however, one study suggested that MIEN1 might cooperate with ΔNp73 to enhance cell proliferation and contribute to cisplatin resistance in ovarian cancer cells [8]. Our [^3^H]thymidine incorporation, EdU staining proliferation, and cloning assays revealed that MIEN1 enhanced cell proliferation in prostate carcinoma cells in vitro. A further xenograft animal study confirmed that MIEN1 knockdown blocked cell growth in vivo (Figure 3). Our study showed that overexpression of MIEN1 enhanced resistance to cisplatin-induced apoptosis and cell viability in prostate carcinoma PC-3 cells (Figure 4). Our research collectively showed that MIEN1 not only increased in vitro and in vivo cell growth but also exhibited anti-apoptotic effects in cells of prostate carcinoma.

MIEN1 is known to play a significant part in preserving the plasticity of the dynamic membrane-associated actin cytoskeleton, resulting in a rise in breast cancer cell motility [23,24]. The present study revealed that overexpression of MIEN1 enhanced the invasion and migration ability of prostate carcinoma cells (Figure 5). The results of F-actin staining from the present study indicated that MIEN1 modulated the epithelial-to-mesenchymal transition (EMT) marker, which enhanced F-actin staining intensity at the leading edge of the cell (Appendix A). These findings are in agreement with other studies which showed that MIEN1 blocked the expression of E-cadherin and stabilized polymers of F-actin [13,24]. 

IL-6 is a multifunctional cytokine known to participate in the malignant progression of prostate cancer [18]. Previous studies confirmed that both IL-6 and IL-6 receptor are expressed in PC-3 cells, and the inhibition of IL-6 gene expression attenuated PC-3 cell proliferation [25,26]. In this research, we discovered that IL-6 is the downstream gene of MIEN1, although the accurate modulation mechanism of IL-6 by MIEN1 is not well understood. However, our study showed that MIEN1 upregulated IL-6 via the Akt signal pathway, while MK2206, an Akt inhibitor, blocked the activation of MIEN1 on IL-6 gene expression (Figure 6). Other studies revealed that upregulation of IL-6 expression in human epithelial colorectal adenocarcinoma (Caco-2) cells and vascular endothelial cells may be possible via the Phosphoinositide 3-kinases (PI3K)/Akt pathway [20,27]. Interestingly, our study also found that IL-6 upregulated MIEN1 expression. Apparently, the present study demonstrated that MIEN1 is upregulated by NF-ĸB signals (Figure 2). Moreover, other studies have demonstrated that IL-6 induces NF-ĸB signals in human intestine epithelial (Caco2-BEE) cells and prostate carcinoma cells [28,29]. Based on the above results, we can conclude that there is a positive regulatory feedback loop for gene expression between IL-6 and MIEN1 via Akt/NF-κB signals. NDRG1 has been widely deemed a tumor suppressor gene in numerous types of cancers, including prostate cancer [15,19,30,31]. For the first time, our research showed that NDRG1 gene expression was downregulated by MIEN1. MIEN1 also can affect NDRG1 gene expression in prostate carcinoma cells via the Akt signal pathways (Figure 7). Interestingly, an early study indicated that NDRG1 downregulated Akt phosphorylation in prostate carcinoma cells [32]. Further research is still required on the precise mechanistic interactions among MIEN1, the Akt-centered signaling network, IL-6, and NDRG1. 

## 4. Materials and Methods

### 4.1. Cell Culture and Chemicals

The prostate cell lines (PZ-HPV-7, CA-HPV-10, LNCaP, PC-3, and DU145) were obtained from BCRC (Hsinchu, Taiwan) and maintained as described previously [33]. Fetal calf serum (FCS) was obtained from HyClone (Logan, UT, USA), and RPMI 1640 media was obtained from Invitrogen (Carlsbad, CA, USA). MK-2206-2HCl and JSH-23 were obtained from Selleck Chemicals LLC (Houston, TX, USA). Cisplatin was purchased from Sigma (St. Louis, MO, USA). 

### 4.2. Immunohistochemical Assays

The human prostate tissue array was obtained from SuperBioChip Laboratories (Cat no: CA; Seoul, Koreas). The pathologic staging and grading of tumors was performed based on the manufacturer’s data sheet. The tissue sections were stained using Bond-Max autostainer with a primary antibody of human MIEN1 (TA504625, OriGene Technologies, Rockville, MD, USA) and detection kit (Bond Polymer Refine Detection DS9800) as described previously [34]. An image was captured and the intensity score (1 = weak; 2 = medium; 3 = intense) was analyzed. 

### 4.3. Expression Vector Constructs and Stable Transfection

The IκB expression vector (pCMV-IκBα) and NIK (MAP3K14) expression vector (pcDNA-NIK) were cloned and transiently transfected into PC-3 cells as described previously [26]. The full-length human MIEN1 expression vector (HG16410-UT) and controlled pCMV3 expression vector were purchased from Sino Biologic Inc. (Beijing, China). Electroporation was conducted using an ECM 830 Square Wave Electroporation System as described previously [34]. Transfected cells (PC-MIEN1 and DU-MIEN1) were selected by using 100 μg/mL of hygromycin (Sigma-Aldrich Co.). To construct mock-transfected cells, PC-DNA and DU-DNA, the cells were transfected with a controlled pCMV3 expression vector and selected clonally in the same manner as described above.

### 4.4. Knockdown of MIEN1

LNCaP and PC-3 cells were transducted with control shRNA lentiviral particles-A (Sc-108080) or MIEN1 shRNA (h) lentiviral particles (sc-72767-V). Two days after transduction, the cells (LN_shCOL, LN_shMIEN1, PC_shCOL, and PC_shMIEN1) were selected by incubation with 10 μg/mL puromycin dihydrochloride for at least another five generations. 

### 4.5. Immunoblot Assays

Equal amounts of whole-cell lysate were loaded onto a 10% or 12% sodium dodecyl sulfate-polyacrylamide gel and assayed by enhancing the chemiluminescence as described by the manufacturer (PerkinElmer Inc., Waltham, MA, USA). The blotting membranes were probed with antiserum of NIK, IĸBα, Akt, phospho-Akt^S473^, Slug (Cell Signaling Technology, Danvers, MA, USA), MMP9, Snail (Abcam, Cambridge, MA, USA), MIEN1 (OriGene Technologies), NDRG1 (Invitrogen Thermo Fisher Scientific Inc., Waltham, MA, USA), E-cadherin (Santa Cruz Biotechnology, Santa Cruz, CA, USA), N-cadherin (Abgent, San Diego, CA, USA), or β-actin (Merck Millipore, Burlington, MA, USA). The intensities of different bands were analyzed using the GeneTools of ChemiGenius (Syngene, Cambridge, UK). 

### 4.6. Real-Time Reverse Transcriptase Polymerase Chain Reaction (RT-qPCR)

The total RNA was isolated using Trizol reagent, and the cDNA was synthesized using the Superscript III pre-amplification system (Invitrogen) as described previously [34]. PCR probes for human MIEN1 (Hs01004335_g1), IL-6 (Hs00985639_m1), PSA (Hs02576345), NDRG1 (Hs00608387_m1), β-actin (Hs01060665_g1), E-cadherin (Hs01023894_m1), N-cadherin (Hs00169953_m1), Snail (Hs00195591P_m1), and Slug (Hs00161904_m1) were purchased from Applied Biosystems (Foster City, CA, USA). Real-time polymerase chain reactions were performed using a CFX Connect Real-Time PCR system (Bio-Rad Laboratories, Foster city, CA, USA), and the mean cycle threshold (C*_t_*) values were calculated for internal control and target genes as described in detail previously [35]. 

### 4.7. Thymidine Incorporation Assays

Cell proliferation was measured using ^3^H-thymidine incorporation, as described previously [36].

### 4.8. EdU Staining Proliferation Assay 

Cells (5 × 10^5^) were cultured in serum-free medium for 24 h. After another 48 h incubation with 10% serum medium, the cells were incubated with EdU (5-ethynyl-2′-deoxyuridine; 10 μM) for a further 2 h. Then, the cells were collected and analyzed using Click-iT EdU Flow Cytometry Assay Kits (Thermo Fisher Scientific Inc.) as described by the manufacturer. The EdU fluorescence of cells was detected using an Attune N × T acoustic focusing cytometer (Thermo Fisher Scientific Inc.). 

### 4.9. MTS Assay

Cells (3 × 10^3^) were seeded onto each well of a 96-well plate in RPMI 1640 medium with 10% FCS for 48 h. MTS dye was added, and the plates were read 2 h later at absorbance of 490 nm.

### 4.10. Colony Formation Assay

The cells were cultured in a 6-well plate (200 cells/well) for 24 h. After the supernatant was removed, the cells continued to incubate in RPMI medium with 10% FCS for another 7 days. The colonies were fixated and counted manually after staining with 0.5% crystal violet as described previously [37].

### 4.11. Annexin V-FITC Apoptosis Detection

The cell pellets were harvested after cells were treated with or without cisplatin for 24 h. Apoptosis detection and quantification were performed after treatments with Annexin V-FITC and propidium iodide (PI) (BioVision Inc, Milpitas, CA, USA) for 5–10 min by using the BD FACSCalibur (BD Biosciences, Bedford, MA, USA), as described previously [38]. 

### 4.12. Matrigel Invasion Assay

Cells that invaded to the other side of a transmembrane coated with Matrigel (BD Biosciences; Bedford, MA, USA) were fixed with 4% paraformaldehyde and then stained with 0.1% crystal violet solution for 10 min. The quantity of cells that invaded the Matrigel was recorded microscopically (IX71, Olympus, Tokyo, Japan) as described previously [36].

### 4.13. Cell Migration Assay

The cell sheets were wounded with a plastic pipette tip when the cells formed a confluent monolayer. The wound closure (the gap width) was photographed under an optical microscope with a digital camera during the indicated times. Quantification of the cell migration was determined within a defined area using the Image J program as described previously [38]. 

### 4.14. Xenograft Animal Model

The performance of animal studies was approved by the Chang Gung University Animal Research Committee (CGU106-157). The male nude mice (BALB/cAnN-Foxn1) were obtained from the animal center of the National Science Council, Taiwan. The mice were anesthetized intraperitoneally, and 3 × 10^6^ cell/100 μL cells were injected subcutaneously on the lateral back in close proximity to the shoulder of each mouse. The growth of the xenograft was recorded by vernier caliper measurement on the indicated days. The tumor volume was determined by the formula Volume = π/6 × W × D^2^, as described previously [34]. 

### 4.15. F-Actin Staining 

Cells were seeded onto glass-bottom culture dishes (MatTek, Ashland, MD, USA), fixed with 3.7% formaldehyde, permeabilized with 0.1% Triton X-100, and blocked in 1% bovine serum albumin (BSA) at room temperature. The F-actin protein expression was revealed by incubation with Texas Red X-Phalloidin (Invitrogen), and immunofluorescence was examined using a confocal microscope (LSM510 Meta, Zeiss, Oberkochen, Germany) as described previously [34].

### 4.16. IL-6 ELISA

The IL-6 levels in the conditioned media were measured and adjusted using the concentrations of proteins in the whole-cell extract as described previously [25]. 

### 4.17. Reporter Vector and Reporter Assays

The NF-κB and human NDRG1 reporter vectors were constructed as described previously [20,29]. The 5′-DNA fragment (-1 to -4573) of the human MIEN1 gene according to the sequence from GenBank (ENSG00000141741) was synthesized by Invitrogen and cloned into the pbGL3 reporter vector (Promega Biosciences, Madison, WI, USA) with *Hind III* sites. Cells were cotransfected transiently with reporter vector and expression vector as indicated using the X-tremeGene HP DNA transfection reagent (Roche Diagnostics GmbH, Mannheim, Germany) as described previously [38]. The luciferase activity was adjusted for transfection efficiency using the normalization control plasmid pCMVSPORTβgal.

### 4.18. Statistical Analysis

All the results are expressed as the mean ± standard error (SE). The statistical significance was determined by one-way ANOVA and Student’s *t*-test using SigmaPlot 10.0 (SPSS Inc, Chicago, IL, USA). 

## 5. Conclusions

These studies indicate that MIEN1 is abundant in advanced prostate cancer and support a role in cancer progression. Overexpression of MIEN1 may contribute to cell proliferation, migration, invasion, tumorigenesis, and cisplatin resistance in prostate carcinoma cells. MIEN1 is a downstream target of NF-κB signaling which facilitates IL-6 induction and suppresses NDRG1 expression via the Akt signal pathway. Additionally, IL-6 modulated a positive feedback loop in activating MIEN1 expression, possibly through NF-κB signaling. Collectively, our study suggests that MIEN1 is the oncogene in the human prostate.

## Figures and Tables

**Figure 1 cancers-11-01486-f001:**
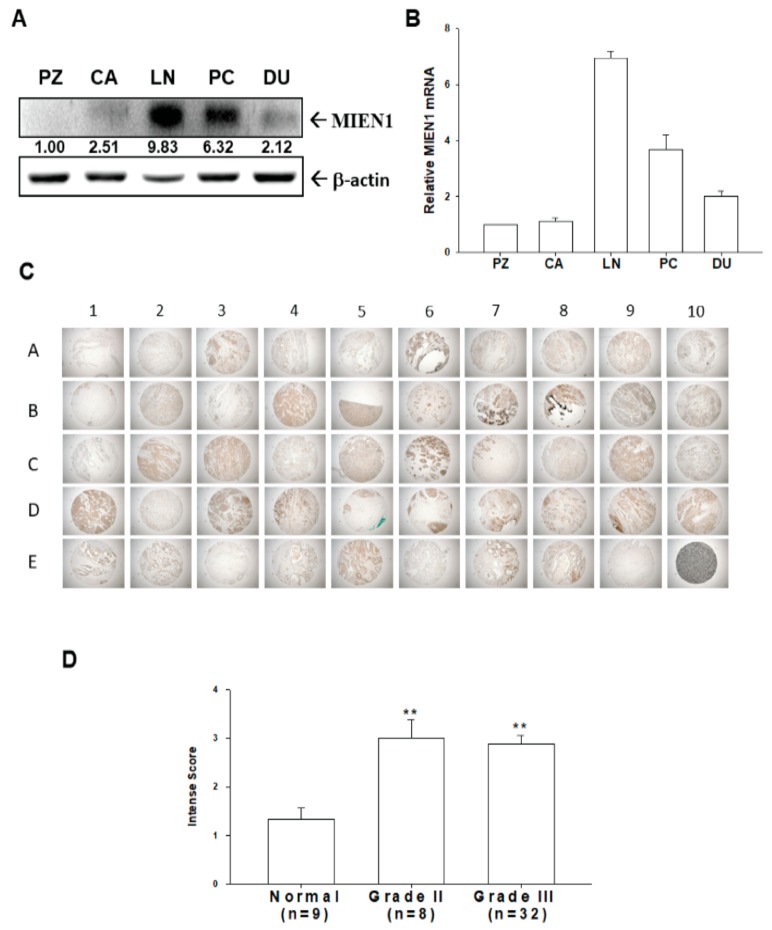
Expression of migration and invasion enhancer 1 (MIEN1) in human prostate carcinoma cells and prostate tissues. The expression levels of MIEN1 in prostate carcinoma cells (PZ: PZ-HPV-7; CA: CA-HPV-10; LN: LNCaP; PC: PC-3; DU: DU145) were determined by (**A**) immunoblotting and (**B**) RT-qPCR. The number indicates the ratio of MIEN1/β-Actin in relation to PZ-HPV-7 cells. (**C**) Immunohistochemical staining for MIEN1 in a human prostate tissue array with normal and prostate cancer tissues (grades II and III). (**D**) The intense scores of MIEN1 immunostaining in normal (*n* = 9) or prostate cancer tissues (grade II, *n* = 8; grade III, *n* = 32). ** *p* < 0.01.

**Figure 2 cancers-11-01486-f002:**
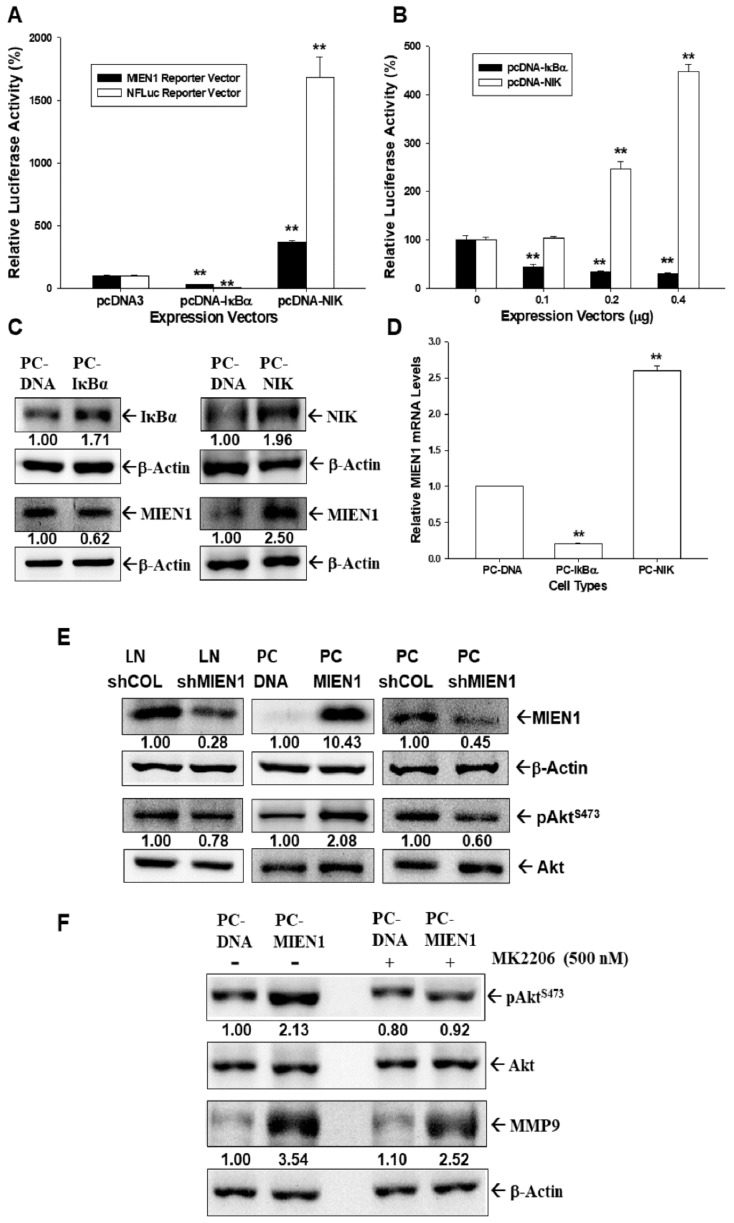
Modulation of migration and invasion enhancer 1 (MIEN1) by NF-κB signaling in prostate carcinoma cells. (**A**) Relative luciferase activity of NF-κB (NFLuc) and MIEN1 reporter vectors cotransfected with NF-κB inhibitor α (IκBκ) or NF-κB induced kinase (NIK) expression vectors as indicated in PC-3 cells. (**B**) Relative luciferase activity of MIEN1 reporter vector after cotransfection with various dosages of NIK or IκBα expression vectors. Data are presented as mean percentage ± standard error (SE) (*n* = 6) of the luciferase activity in relation to the vehicle-treated group (** *p* < 0.01). (**C**) Protein levels of IκBα, NIK, and MIEN1 after ectopic IκBα or NIK overexpression in PC3 cells. (**D**) Relative mRNA levels of MIEN1 in IκBα- or NIK-overexpressing PC-3 cells (± SE, *n* = 3). (**E**) Protein levels of MIEN1, Akt, and pAkt^S473^ in PC-MIEN1, PC_shMIEN1, and LN_shMIEN1 cells in comparison to the mock-transfected control cells (PC-DNA, PC_shCOL, and LN_shCOL). (**F**) Protein levels of Akt, pAkt^S473^, and matrix metallopeptidase 9 (MMP9) in PC-DNA and PC-MIEN1 cells after treatment with or without MK2206 (500 nM) for 24 h. The number indicates the ratio of target gene/β-Actin or pAkt^S473^/Akt in relation to PC-DNA, LN-shCOL, or PC-shCOL cells.

**Figure 3 cancers-11-01486-f003:**
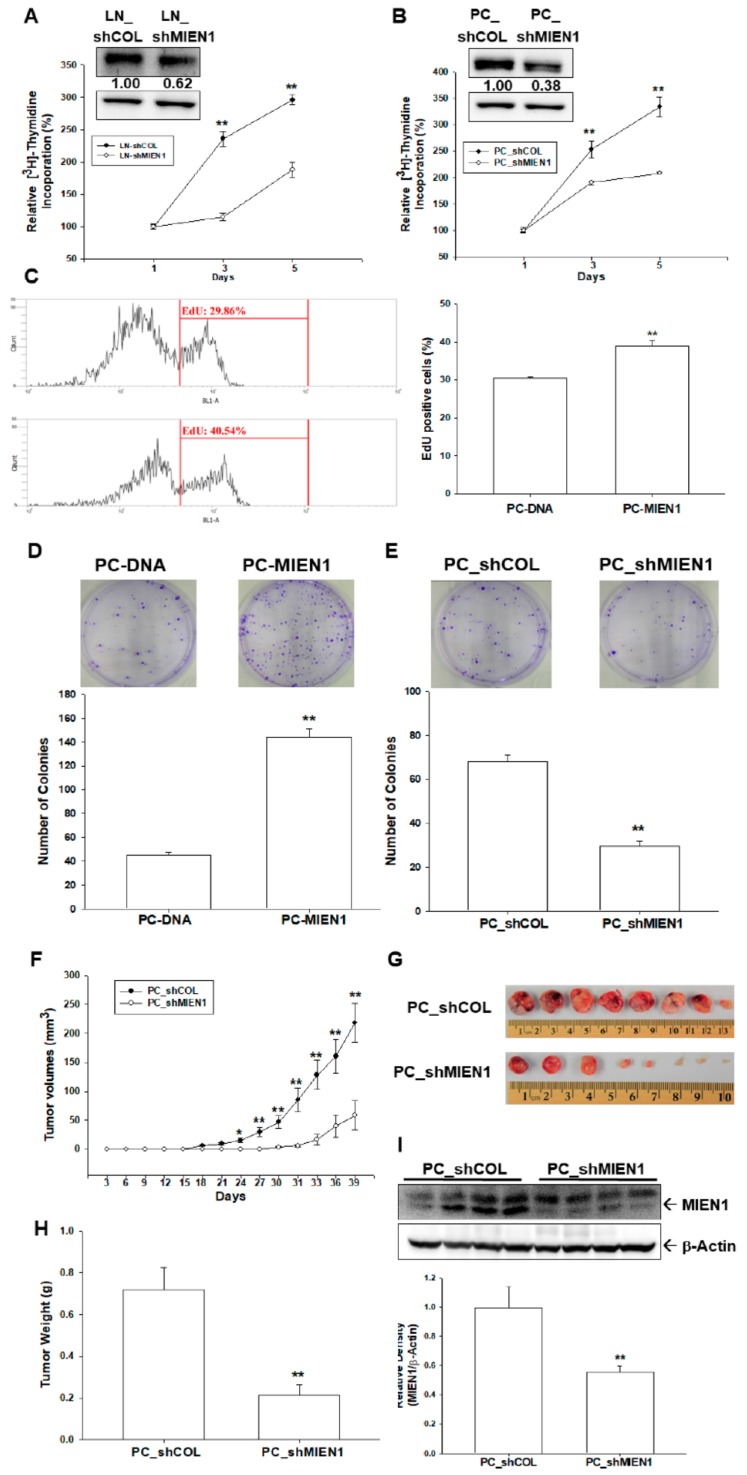
Effects of migration and invasion enhancer 1 (MIEN1) on cell proliferation and tumorigenesis in prostate carcinoma cells. Cell proliferation of LNCaP (**A**) and PC-3 (**B**) cells after MIEN1 knockdown was determined by ^3^H-thymidine incorporation assays. The number indicates the ratio of MIEN1/β-Actin in relation to LN-shCOL or PC-shCOL cells. (**C**) The proliferation ability of PC-DNA and PC-MIEN1 cells was determined by flow cytometry via Click-iT EdU flow cytometry assay (± SE, *n* = 4). Cell proliferation with MIEN1 overexpression (**D**) and MIEN1 knockdown (**E**) in PC-3 cells was determined by colony formation assay (± standard error (SE), *n* = 6). Four-week-old male athymic nude mice were divided randomly into two groups. PC_shCOL or PC_shMIEN1 cells (3 × 10^6^) were injected subcutaneously into the dorsal area of the mice (*n* = 8). (**F**) The tumor volumes were measured every 3 days during a period of 39 days. The tumor size (**G**) and tumor weight (**H**) were measured after sacrifice. (**I**) The quantitative analysis was done by determining the intensities of MIEN1 and β-Actin. Data are presented as the fold induction (± SE, *n* = 4) of the relative density of the MIEN1/ β-Actin in relation to the PC_shCOL group. * *p* < 0.05, ** *p* < 0.01.

**Figure 4 cancers-11-01486-f004:**
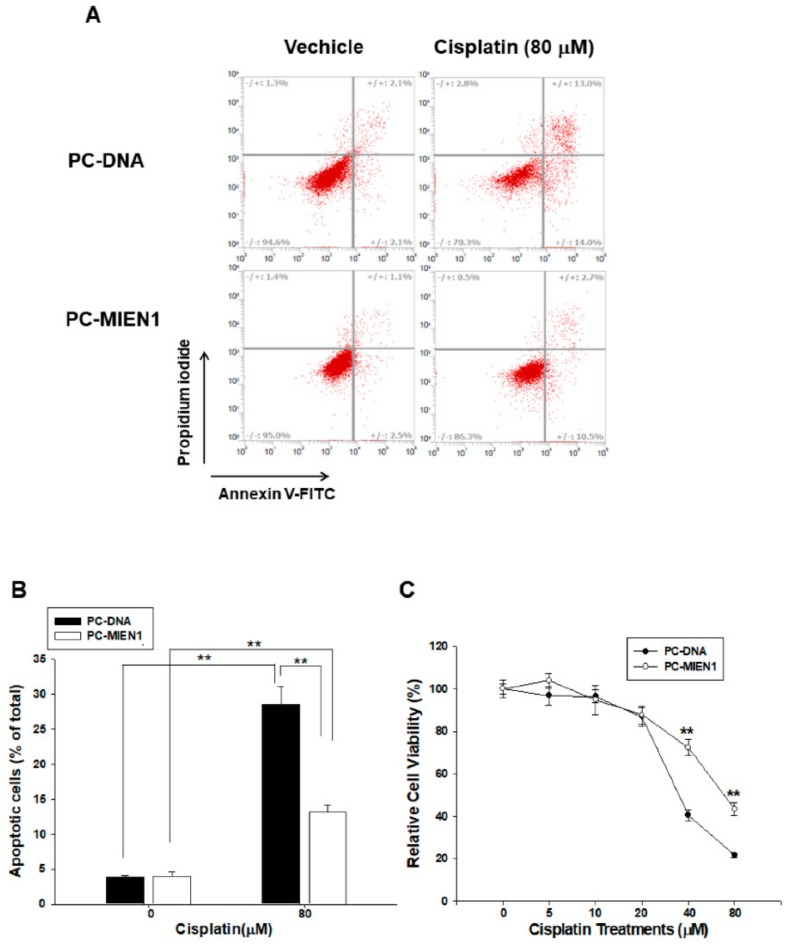
Overexpression of migration and invasion enhancer 1 (MIEN1) attenuates cisplatin-induced cell apoptosis in PC-3 cells. (**A**) Cells (PC-DNA and PC-MIEN1) were treated with or without 80 μM cisplatin for 24 h. The fluorescence intensity for Annexin V-FITC in conjunction with PI staining was determined by flow cytometry (± standard error (SE), *n* = 4). (**B**) Data are presented as the percentage of apoptotic cells after cisplatin treatments. (**C**) Cell viability of PC-DNA and PC-MIEN1 cells after treatment with various dosages of cisplatin as indicated for 48 h (± SE, *n* = 8). ** *p* < 0.01.

**Figure 5 cancers-11-01486-f005:**
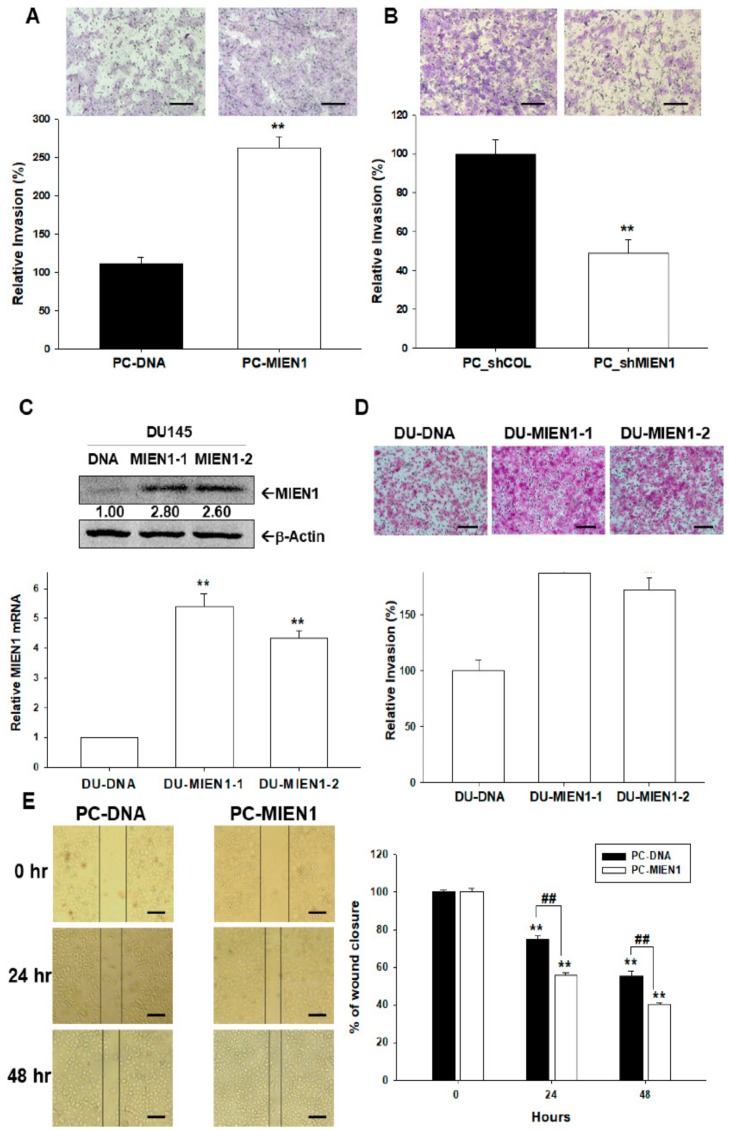
Effects of migration and invasion enhancer 1 (MIEN1) on cell migration and invasion in prostate carcinoma cells. (**A**) Invasion ability of MIEN1 overexpression in PC-MIEN1 or MIEN1 knockdown (PC_shMIEN1) (**B**) relative to that of mock-transfected cells (PC-DNA or PC_shCOL) determined by Matrigel invasion assays (± standard error (SE), *n* = 3). (**C**) The expression of MIEN1 in DU145 (DU-MIEN1-1, DU-MIEN1-2) cells was determined by immunoblot (C, top) and RT-qPCR (C, bottom) assays. The number indicates the ratio of MIEN1/β-Actin in relation to DU-DNA cells. (**D**) Invasion ability of DU-MIEN1-1 and DUMIEN1-2 cells in relation to mock-transfected (DU-DNA) cells (± SE, *n* = 3). The scale bar is 200 μm (**E**) The migration ability of PC-DNA and PC-MIEN1 cells was determined by wound healing assays at the indicated times (± SE; *n* = 4). The scale bar is 100 μm. **^,##^
*p* < 0.01.

**Figure 6 cancers-11-01486-f006:**
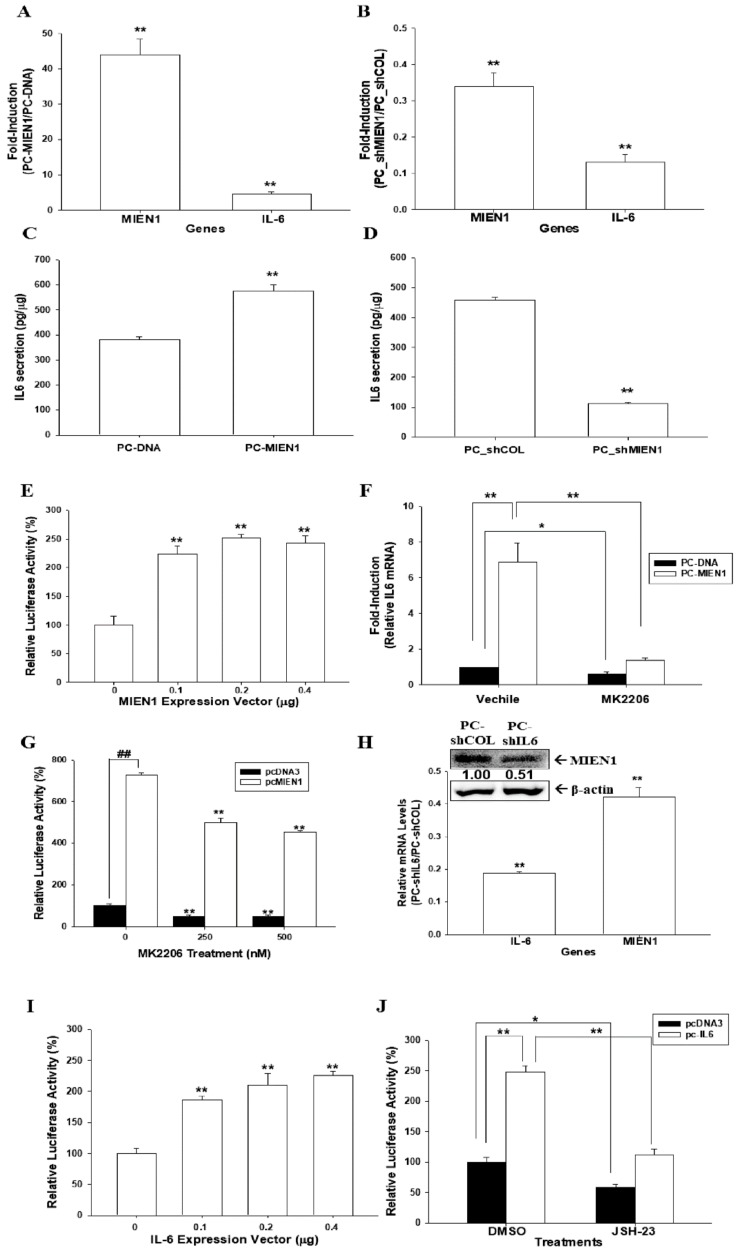
Modulation between migration and invasion enhancer 1 (MIEN1) and interleukin-6 (IL-6) in prostate carcinoma PC-3 cells. Relative expression of MIEN1 and IL-6 after ectopic MIEN1 overexpression (**A**) or MIEN1 knockdown (**B**) in PC3 cells as determined by RT-qPCR assays. Data are presented as the mean fold induction of the mRNA levels (± standard error (SE), *n* = 3) relative to the mock-transfected groups PC-DNA and PC_shCOL, respectively. Secretion of IL-6 in MIEN1-overexpressing (**C**) or MIEN1-knockdown (**D**) PC3 cells. (**E**) Relative luciferase activity of IL-6 reporter vector after cotransfection with various dosages of MIEN1 expression vector (± SE, *n* = 6). (**F**) Relative expression of IL-6 in MIEN1-overexpressing PC3 cells treated with/without MK2206 as determined by RT-qPCR assays (± SE, *n* = 3). (**G**) The relative luciferase activity of IL-6 reporter vectors after transient overexpression of MIEN1 expression vector and treatment with/without MK2206 (± SE, *n* = 6). (**H**) Relative expression of IL-6 and MIEN1 after MIEN1 knockdown in PC3 cells determined by immunoblot and RT-qPCR assays. The number indicates the ratio of MIEN1/β-Actin in relation to PC-shCOL cells. (**I**) Relative luciferase activity of MIEN1 reporter vector after cotransfection with various dosages of IL-6 expression vector. (**J**) The relative luciferase activity of MIEN1 reporter vectors after transient overexpression of IL-6 expression vector and treatment with/without JSH-23 (± SE, *n* = 6). * *p* < 0.05, **^,##^
*p* < 0.01.

**Figure 7 cancers-11-01486-f007:**
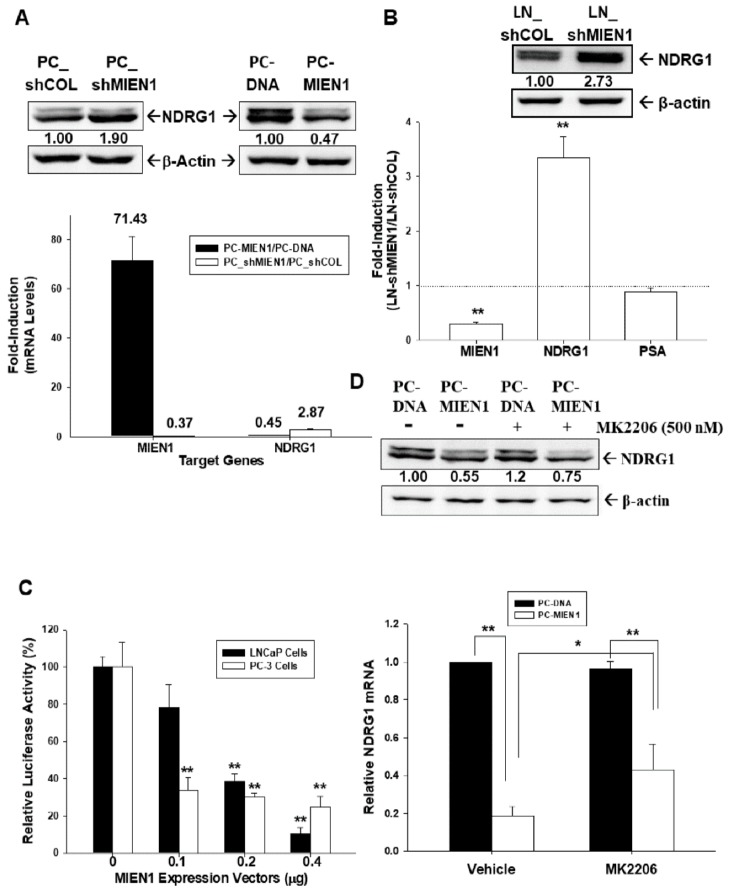
Migration and invasion enhancer 1 (MIEN1) downregulates N-myc downstream regulated 1 (NDRG1) gene expression in prostate carcinoma cells. (**A**) Relative expression of NDRG1 in MIEN1-overexpressing or MIEN1-knockdown PC-3 cells determined by immunoblot (top) and RT-qPCR (bottom; ± standard error (SE), *n* = 3; the number indicates the fold induction) assays. (**B**) Relative fold induction of MIEN1, NDRG1, or prostate specific antigen (PSA) gene expressions in MIEN1-knockdown LNCaP cells determined by immunoblot (top) and RT-qPCR (bottom) assays (± SE, *n* = 3). The number indicates the ratio of NDRG1/β-Actin in relation to PC-DNA, LN-shCOL, or PC-shCOL cells. (**C**) The relative luciferase activity of NDRG1 reporter vector after cotransfection with various dosages of MIEN1 expression vector in LNCaP and PC3 cells (± SE, *n* = 6). (**D**) Relative expression of NDRG1 in PC-MIEN1 cells with/without MK2206 determined by immunoblot (top; the number indicates the ratio of MIEN1/β-Actin in relation to vehicle-treated PC-DNA cells) and RT-qPCR (bottom; ± SE, *n* = 3) assays. * *p* < 0.05, ** *p* < 0.01.

## Data Availability

Please contact the corresponding authors for all data requests.

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
