# Peer review of "Migration and Invasion Enhancer 1 Is an NF-ĸB-Inducing Gene Enhancing the Cell Proliferation and Invasion Ability of Human Prostate Carcinoma Cells In Vitro and In Vivo"

_cancers, 2019, doi:10.3390/cancers11101486_

Round 1

Reviewer 1 Report

In the study entitled “Migration and invasion enhancer 1 is a NF-kB-inducing gene enhancing cell proliferation and invasion ability of human prostate carcinoma cells in vitro and in vivo”, the authors investigated the crucial role played by MIEN1 in prostate cancer progression. Although the issue of the study is interesting and experiments well carried out, it lacks the main focus and it reports many data that could be addressed in more depth in another manuscript e.g. MIEN1 antiapoptic effect and its correlation with drug-resistance. The study should be totally revised above all the introduction and the discussion. The conclusion must be rewritten. Extensive editing of English language and style required.

I would like to suggest the authors to address the following points to improve the manuscript.

Major Revision

The references 1, 2, 3, 4 are missing

The reference 12 is wrong after the sentence reported in lines 46-48. The appropriate reference is 16.

Similarly, ref. 6 is wrong and the appropriate reference is 10

The appropriate reference in the sentence reported from line 48 to line 51 is not 13 but Rajendiran S, Parwani AV, Hare RJ, Dasgupta S, Roby RK, Vishwanatha JK. MicroRNA-940 suppresses prostate cancer migration and invasion by regulating MIEN1. Mol Cancer. 2014 Nov 19;13:250. doi: 10.1186/1476-4598-13-250. This reference is not reported in the manuscript.

In the introduction the importance of IL-6 and NDGR1 in this study are not described

The relative protein levels by densitometric analysis should be reported for each western blot as shown in figure 7

In particular, the author must report the data expressed as mean /SD (or ES) of MIEN1 levels of 16 xenografts analyzed

The experiments performed to demonstrate the anti-apoptotic role of MIEN1 is too little exhaustive.

The author reported that MIEN1 upregulates IL-6 and in turn IL-6 upregulates MIEN1, this issue should be addressed better in the results and in the discussion. It is not very clear. In particular, the role of this loop in the migration and invasiveness of prostate cancer cells should be better discuss.

In Figure S1 the results reported in the graphic are not consistent with western blot in particular slug levels.

Reviewer 2 Report

The authors describes MIEM1 activate prostate cancer cell migration, invasion, proliferation and tumorigenesis by NFkB signaling activation. The findings are interesting and important, however the manuscript lacks consistency and evidence. Also, there are so many miss-spelling and problems on document format.

IkBa and NIK regulate MIEM1 expression in protein level as shown in figure 2C. There are not much difference in MIEM1 protein level between PC-DNA and PC-IkBa in immunoblot analysis. However, as shown in 2D, mRNA levels of MIEM1 are much different by modifying NFkB pathway. Please discuss about this discrepancy. They used several prostate cancer cell lines. I think the reason is to show the finding is universal in prostate cancer. However, the data presented in the manuscript are lack of consistency. (for example: I cannot found LN-MIEN1 cell and DU-shMIEM1 cell in the figures) Please show the data from another cell lines to strengthen the authors claims. Figure 7A bottom; mRNA levels of PSA should be presented as 7B. Figure 7C; This is indirect evidence that MIEM expression level influence NDRG1 reporter activity. It is better to show the expression level of MIEM1 directly. Most importantly, they showed the link between MIEM1 and NFkB signaling and IL6 induction, however, they did not show the direct link between NFkB signaling and prostate cancer cell proliferation, migration, invasion and tumorigenesis. The references do not totally link the citation number on the main manuscript. I recommend to check manuscript format again.

Round 2

Reviewer 1 Report

The revised version of the manuscript has drastically improved, therefore it is suitable to be published in Cancers

Reviewer 2 Report

I think the authors satisfactory respond to my points.